# Beyond the game: How social interaction and emotional gratification drive Chinese sports podcast engagement

Fangni Li *

School of International Studies, Communication university of China, Beijing, China

* fangni.li@foxmail.com

## Abstract

The rapid growth of podcasting platforms has transformed digital media consumption, offering diverse content tailored to user preferences. The present research aims to identify the needs associated with engaging with sports podcasts on the Himalaya platform while exploring cognitive, emotional, and social satisfactions. Quantitative and qualitative data collection techniques used the Uses and Gratifications Theory (UGT) in the present research, including semantic network analysis, co-word clustering, and high-frequency words, to analyze the data collected from 13,692 comments of five popular sports podcast channels. The study shows that information seeking is the primary reason listeners tune in: they want game breakdowns and insiders' opinions. This is further reinforced by self-efficacy and media convenience, which are important factors in the platform mandate of enabling users and assimilating content into their daily lives. First, the paper adds to scholarship by employing the UGT in the context of sports podcasts in China. Secondly, it provides valuable recommendations for practical use by content creators and platform developers. Overall, the results show the versatility of podcast consumption, although restrictions concerning demographic categorization and comparison with other media forms signal to undertake future studies. Such insights are relevant to improving and encouraging user interactions and how the platform delivers podcasts.

## 1. Introduction

The evolution of digital media technologies has significantly transformed how audiences consume information and entertainment, with podcasting emerging as one of the most popular formats globally [1,2]. In China, Ximalaya App, announced by Ximalaya Inc. in 2012, has become a leading player in the podcasting industry. It provides a bottomless well of audio across any category, including education, lifestyle, and sports [3]. One of the concept's strengths is complex recommendation tools and unusual active elements for increasing user engagement. Himalaya has easily

**Data availability statement:** The dataset analyzed in this study consisted of 13,692 publicly posted user comments collected from five sports podcast channels on the Himalaya platform between August and October 2024. These comments were accessed using a custom Python scraper that complied with the platform's publicly available Terms of Service at the time of collection. Unfortunately, due to platform restrictions and data retention policies, the raw dataset is no longer accessible. However, the study's full methodology, including scraping scripts, processing workflow, and analytical procedures, is detailed in the Methods section, enabling reproducibility. No personally identifiable information (PII) was collected, and all data were anonymized during analysis.

**Funding:** The author(s) declare that financial support was received for the research and/or publication of this article. This research was supported by the Fundamental Research Funds for the Central Universities (Grant number: CUC25GG13 to F.L.).

**Competing interests:** The authors have declared that no competing interests exist.

workable interconnectivity through other digital environments like social media and video-sharing platforms, adding to its desirability notably among young audiences and high-income users [3,4]. Listeners have embraced sports podcasts in the Himalaya App, which include a variety of live coverage, discussions, analysis, player and team interviews, and fans' real-time chats. Each program offers the audience completely interactive experiences through stories of rivalry, companionship, and fair play [5]. Compared to conventional media, podcasts on Himalaya enable users to listen to programs conveniently, during commuting or otherwise. Despite the increasing prominence of such services as Himalaya, academic research on the function and efficacy of such platforms in promoting involvement in physical activity, including in China's distinctive social environment, is still insufficient [3,5].

In this digital age, there is more of a scatter of audiences, where they shift to media forms that allow one to do something else simultaneously, for example, podcasts. Internationally, this change has occurred when other social media platforms such as YouTube, TikTok, and others, heavily based on videos, are on the rise with personalized content and heavily weighted on imagery [6,7]. On the other hand, podcasts present fully formed auditory content, made for attention, focus, and listening, which is quite the opposite in an era where the new normal is the 'nap and swipe' culture, raising questions about what aspects of the broader concept of podcasts and podcasting are making them popular now – and why sports-related content, which is not only timely but heavy on analysis, is so popular [8]. The podcasting industry in China is still relatively young compared to the market's more developed podcasting industries, such as in the United States or Europe. Although an advanced literature review has discussed podcast consumption behavior in nations like the United States, a void regarding Chinese sports podcast listener motives has been identified [9]. Moreover, the Himalaya, for instance, stands and functions in a different cultural and technological environment than defined by the two aforementioned studies, shaping the audiences' behaviors. Therefore, this research aims to fill the gap by investigating the flow-based motivation of sports podcasts in China with Himalaya as the focal case [10].

Consequently, the Himalaya App as an object of the study can be valid given that the platform is the key audio content provider in China, and the availability of the sports topic podcasts allows for exploring the nature of its consumption. This inventory contains a wide range of sports programs that can meet the interests of any listener, including analyses of the games and stimulating conversations [11,12]. In addition, embracing user comments, interactive tools, and engine-based suggestions creates a bank of information to analyze audience engagement. Utilizing the Uses and Gratifications Theory (UGT), this paper aims to find out how sports podcasts satisfy the usage needs of audiences [13–15]. This present study furthers the understanding of podcast listenership specifically by identifying key determinants of engagement and tailoring the present analysis to those aspects of listener behavior. The research also adopts text analysis features, including semantic network analysis, to analyze user comments on podcasts involving sports, enabling a systematically find audience motivations and simultaneously reveal the strengths and weaknesses

of textual automation analysis in media studies, providing helpful ideas for academic researchers developing theories and hypotheses and media specialists creating and implementing media content and promotion plans conducive to different users' needs.

Despite the exponential growth of podcast consumption in China, academic discourse on this subject remains limited, especially concerning sports podcasts. Most of the earlier research in podcasting is based on Western countries, and knowledge acquired from these studies may not fit well into the Chinese socio-cultural and technological context [16,17]. Secondly, while there is increasing attention to text analysis using co-word clustering and semantic network analysis applied in other contexts, it has not been prominently used to analyze the potential users' motivations for podcasting. This research addresses these gaps by conducting a focused analysis of sports podcasts on the Himalaya App, shedding light on how Chinese audiences engage with this medium [18]. This research contributes to the theoretical development of audience behavior by developing a motivational model for listening to sports podcasts. In addition, it adds to a growing body of literature on how media, technology, and culture influence and inform content use in non-western contexts.

The contributions of this research are both theoretical and practical. Theoretically, it establishes an extensive motivational model for Chinese sports podcast audiences based on UGT. This model extends the understanding of why podcasts are consumed by considering the fulfillment of cognitive, affective, and social needs. Further, the study provides a textual analysis and evaluation of the current status of text analysis methods and provides new methodologies for analyzing user-generated content that is important in media research. On the methodological level, the research offers recommendations to podcast creators, media and platform marketers, and developers. The study provides insight into the content creation and interaction processes by determining the users' radio frequency spectrum use incentive factors, including information, social interaction, and psychological self-efficacy. Moreover, the results shed light on the cultural and technological factors that may affect Chinese consumers, which will provide directions for customized services according to different users.

This study aims to answer the following central research questions: [1] What cognitive, emotional, and social gratifications can be identified from user-generated comments on Chinese sports podcasts, and how do these gratifications drive audience engagement? [2] How do semantic relationships and thematic clusters in user comments reveal patterns of podcast consumption behaviors on the Himalaya platform? [3] How can the UGT be applied and extended to construct a culturally grounded motivational model of podcast engagement in the Chinese digital media context? These questions are explored through a mixed-methods approach that integrates semantic network analysis, co-word clustering, and word frequency analysis to interpret over 13,000 user comments across five major podcast channels. These research questions frame the study precisely and provide a logical trajectory for the methodological and analytical sections. Furthermore, the research problem is situated in a larger context—the lack of culturally specific media consumption studies in the Chinese digital ecosystem. By clearly defining the study's aim to construct a UGT-based motivational model for podcast engagement in China, the introduction now fulfills its role as a contextual background and a roadmap for the entire manuscript.

## 2. Related review

### 2.1. The global and chinese landscape of podcast research

Podcasting has emerged as one of the most influential formats in contemporary digital media, blending features of radio, on-demand audio, and social networking into an increasingly personalized medium of communication. Internationally, scholarship has documented the evolution of podcasting from its early "do-it-yourself" culture to a professionalized industry that encompasses news, education, and entertainment [7,19]. Sullivan [7] situates podcasts within broader media ecosystems, showing how platform governance and technological infrastructures mediate production and distribution. Similarly, Markman [20] highlighted the hybrid nature of podcasts, which serve as both personal and professional media, with motivations including self-expression, community building, and entrepreneurial goals.

Educational research has also demonstrated the pedagogical applications of podcasts, ranging from higher education [8,21] to motivational tools in blended learning [22] and EFL classrooms [23]. This literature highlights the ability of podcasts to serve as flexible, learner-driven content, paralleling broader digital education trends [17]. The combination of accessibility, intimacy, and narrative storytelling positions podcasts as distinct cultural and social products [14].

In China, podcasts have developed along a unique trajectory. Platforms like Himalaya FM, which launched with significant venture capital backing [12], have rapidly expanded to serve millions of users. UNESCO reports that Himalaya FM has developed a "radio à la carte" model for Chinese audiences, enabling customized access to a diverse range of content, from entertainment to knowledge-driven programming [5]. Domestic scholarship has identified challenges and opportunities in the Chinese podcast industry, including monetization, audience segmentation, and cultural integration [16]. Yang and Wang [3] further contextualize podcasts within the broader ecology of Chinese social media platforms such as Weibo, Douyin, and WeChat, emphasizing the platformization of public discourse.

Despite this expansion, empirical studies on Chinese podcast audiences remain limited. Chan-Olmsted and Wang [9] provided one of the few large-scale examinations, identifying consumption motives such as entertainment, companionship, and learning. However, most extant studies focus on Western audiences [24,25]) or specific educational contexts [26]. Research on Chinese listeners, particularly in sports domains, is scarce. This gap is striking, given the rising popularity of sports podcasts globally and domestically, as well as their role in shaping community identity and media consumption behaviors [3,16].

Sports-focused podcast research outside China highlights the distinctiveness of podcasts as spaces where fans co-construct narratives, participate in communities, and consume both analysis and entertainment [14,27]. However, comparative evidence on how these motivations play out within China's high-context cultural setting is absent. Addressing this absence provides not only empirical novelty but also theoretical leverage for understanding how global media theories, such as UGT, adapt to culturally specific contexts.

## 2.2. Uses and gratifications theory in podcast and media research

The UGT has served as a foundational framework for analyzing why audiences actively engage with media. Originating in mid-20th-century media studies, UGT emphasizes audience agency, arguing that individuals consume media to satisfy cognitive, affective, and social needs [15]. Contemporary applications extend this logic to digital media, where interactive and participatory affordances amplify user-driven motivations [13,28].

In podcast research, UGT has been operationalized to capture motivations ranging from information acquisition and entertainment to companionship and identity formation. Perks et al. [25] developed a Podcast Uses and Gratifications Scale, empirically validating dimensions including learning, social engagement, entertainment, and relaxation. Chan-Olmsted and Wang [9] confirmed similar motives in a large-scale US survey, highlighting the balance between hedonic and utilitarian gratifications. Likewise, Askar and Mellor [24] identified curiosity, leisure, and identity exploration as central drivers among young Arab audiences, underscoring cultural variability.

UGT's explanatory power extends beyond podcasting into adjacent media forms. Dolan et al. [13] applied UGT to social media engagement, demonstrating that gratifications such as entertainment, information, and social interaction significantly predicted the intensity of engagement. In e-commerce, Lim and Ting [15] employed UGT to explain shopping motivations, while Yang and Tasi [29] applied it to podcast features, demonstrating that satisfaction and compatibility mediate users' intention to continue using the service. Collectively, these studies reaffirm UGT's flexibility as a framework across various domains, while also inviting further testing in non-Western contexts.

China represents a critical test case for UGT. Its digital media ecology is shaped by platform-specific governance [3], collectivist cultural norms [30], and linguistic particularities [31]. Research shows that Chinese audiences often frame gratifications through relational and hierarchical concepts, e.g., respect for expertise, group belonging, and ritualized

engagement [5]. In podcast contexts, the frequent invocation of "teacher" to describe hosts exemplifies how cognitive gratifications intertwine with cultural norms of authority and respect [8,24]. This points to an opportunity to enrich UGT by situating it in high-context cultures where emotional resonance and social hierarchy are salient. Thus, while UGT has been extensively validated in Western settings, its cross-cultural adaptability remains underexplored. Applying UGT to Chinese sports podcasting provides a pathway to refine the theory, demonstrating how global constructs are rearticulated in culturally specific forms.

## 2.3. Mixed-methods approaches in digital media and audience research

Methodologically, studying digital media consumption requires approaches that capture both scale and nuance. Mixed methods, which combine quantitative breadth with qualitative depth, have gained increasing traction across various fields [32,33]. The convergent parallel design, in which qualitative and quantitative strands are conducted simultaneously and integrated at the interpretation stage, offers a robust strategy for examining complex motivations.

Within audience studies, mixed-methods research is especially valuable. Quantitative approaches, such as semantic network analysis [18,34] and co-word clustering [35,36], offer systematic methods for detecting patterns in large-scale user-generated content. Word frequency analysis, a foundational content analysis tool [37], identifies salient terms, while advanced computational linguistics extends this to uncover relational structures ([38, 39]).

Qualitative approaches complement these methods by contextualizing meanings, ensuring that cultural and linguistic nuances are not lost in computational abstraction [11]. Triangulation of data sources and methods further enhances validity [40,41]. Recent research demonstrates the strength of integrating text mining with qualitative interpretation. For instancee, Ellahi et al. [18] applied semantic network analysis to investigate service experiences, showing how computational insights benefit from interpretive framing. Malandrino [42] compared qualitative coding with network-based analysis to highlight complementarities in understanding policy discourses. These studies underscore that mixed methods are not only pragmatic but theoretically generative, allowing researchers to identify patterns while also interpreting their cultural and social significance.

In podcast research, however, mixed-methods applications remain limited. Most studies adopt surveys [9,25] or qualitative interviews [14]. Few studies systematically integrate large-scale user-generated data with interpretive analysis [3,4,20]. Given the scale of digital interaction on platforms like Himalaya and the cultural richness embedded in Chinese-language comments, a convergent mixed-methods design is particularly well-suited for this study.

## 2.4. Synthesis and research gap

Existing studies highlight three interrelated insights. First, podcasts are a rapidly growing medium globally and in China, yet empirical research on Chinese sports podcasts is minimal. Existing studies are predominantly based in Western contexts, leaving gaps in understanding how cultural factors influence podcast engagement. Second, while UGT provides a robust framework for analyzing motivations, its application to Chinese audiences remains scarce. Evidence suggests that gratifications such as respect for authority and group belonging may extend UGT's explanatory categories in non-Western settings. Third, methodological innovation is required to capture the complexity of digital media engagement. Mixed methods, particularly those integrating computational text mining with qualitative interpretation, are well-positioned to address this challenge but are underutilized in podcast studies. This study addresses these gaps by employing a convergent mixed-methods design to analyse over 13,000 user comments from China's leading sports podcast platform. By integrating word frequency analysis, semantic networks, and co-word clustering with qualitative interpretation, the research constructs a culturally grounded motivational model of podcast engagement. In doing so, it advances podcast scholarship, extends UGT across cultural frontiers, and demonstrates the methodological value of mixed methods in digital audience research.

## 3. Research methodology

### 3.1. Research design

A mixed methods approach was adopted in the study to thoroughly investigate the motivations of sports podcast listeners on the Himalaya platform. This approach was taken as it balances the depth of quantitative analysis with the width of qualitative insights [26,32]. The theoretical foundation was based on the UGT because, as in the UGT, media users tend to become the actors themselves in choosing and consuming content aimed at a complementary need. The theoretical orientation for this study then justified the focus on audience-driven motivations and aligned the study's goals with extant frameworks in media consumption research [28]. A mixed methods design was even more justified after considering the research problem and its complexity. Podcast listening behaviors are driven by several cognitive, emotional, and social factors and, as such, require integrating multiple analytical techniques. Qualitative methods enabled understanding user comments on a vector, while quantitative methods enabled systematic identification of patterns and relationships in the data [33]. The dual approach is needed to construct a comprehensive motivational model for Chinese sports podcast consumption.

This study employs a convergent mixed-methods design, integrating quantitative text mining and qualitative content interpretation to explore user motivations. The theoretical foundation is anchored in the Uses and Gratifications Theory. Data were sourced from 13,692 user comments posted on five high-traffic sports podcast channels on the Himalaya platform: *Yang Yi Radio*, *Sports Basketball Show*, *Football First Perspective*, *Nan's Talk*, and *CBA Fans Radio*. These channels were purposively sampled based on three criteria: [1] high play and comment volumes, [2] genre diversity (basketball, football, general sports), and [3] active listener engagement over a three-month period (August–October 2024). Comments were extracted using a custom Python web scraper employing Selenium and BeautifulSoup for automated, accurate data retrieval. Post-extraction, the dataset was cleaned using Pandas and NLTK, followed by Chinese word segmentation using the Jieba tokenizer.

### 3.2. Sampling methods

The study employed purposive sampling to select five sports podcast channels on the Himalaya platform as the primary data sources. These channels were selected with popular and high engagement metrics, including subscriber count, play count, and the majority of user comments for channels. Out of all the channels, it was important to select channels with active and diverse audiences to increase the representativeness of the findings [33,43]. This sampling method was justified to ensure that it would capture various listener experiences and motivations. The study filtered in on popular channels with content connected with a big chunk of podcasting audience [44]. The channels included *the Sports Basketball Show*, *Yang Yi Radio*, *Football First Perspective*, *Nan's Talk*, and *CBA Fans Radio*. These channels covered a variety of sports genres, from basketball and football to broader sports commentary, providing a comprehensive dataset for analysis. The decision to use purposive sampling over random sampling was based on the nature of the research question. While random sampling had its merits for generalizability, it was not optimal for the study's objectives, which were to thoroughly investigate specific reasons audiences have for consuming news. For channels, purposive sampling ensured that the Channels that would most likely yield rich and relevant data for the Research focus were included.

### 3.3. Data collection

Data were collected from user comments posted on the selected sports podcast channels over three months. The time range to capture the interaction sample was chosen to surface seasonal variations in sports podcast engagement. A Python-based web scraping tool extracted comments from the Himalaya platform to facilitate efficient, systematic data gathering. User comments was focused to consider authentic time [45]. Unlike survey responses or interview transcripts, user comments gave real-time insight into audience interactions and motivations without the bias from recall or

shape-heightening social desirability effects. Moreover, they were well suited as data sources for exploring how listeners interact with sports podcasts in a naturalistically actual situation [46]. Several preprocessing steps were being done to maintain the quality and validity of the dataset. Links, emojis, and noise symbols were ignored, and redundant comments were removed to ensure the comments did not repeat themselves. A combination of manual screening and automated detection algorithms were used to flag comments that appeared to be either produced by bots or as part of paid campaigns [47,48]. These algorithms analyzed patterns like repetitive phrasing and unusually high posting frequency to detect nonorganic content [49]. About 280,000 words of textual data comprised the final dataset, comprising 13,692 validated comments. Web scraping was justified since it was best suited to deal with large-scale data efficiently while retaining the original structure and content of comments [45]. The link to the study's goal of analyzing user-generated content at scale was perfect for analyzing motivational themes.

Using a web scraping technique based on Python, specifically Selenium and BeautifulSoup, the user comments were retrieved from the Himalaya platform's publicly accessible podcast comment sections. The data were accessible to the public and could be viewed at the time of gathering according to Himalaya's Terms of Service, which allowed academic study that was not for profit. No personally identifying information (PII) was gathered, kept, or used in any way; all data was textual and anonymized. While the original comment dataset is no longer accessible due to data retention constraints, the study's reproducibility is guaranteed by providing a detailed description of all scraping and preprocessing techniques.

### 3.4 Mixed methods design clarification

This study employed a Convergent Parallel Mixed Methods Design, a widely recognized design within the typologies outlined by Creswell and Clark [50]. In this approach, the quantitative and qualitative strands of data collection and analysis are conducted independently yet concurrently, and their results are subsequently integrated to provide a more comprehensive understanding of the research problem. This explicit alignment with the established framework removes ambiguity and ensures the design is both methodologically sound and transparent [51]. The rationale for adopting a convergent parallel design lies in the dual nature of the research objectives. The quantitative strand, which utilized large-scale text mining techniques, was indispensable for mapping broad patterns and trends that would have been impossible to discern through qualitative analysis alone [33]. This provided the study with a robust measure of breadth and generalizability. At the same time, the qualitative strand offered interpretive depth, enabling a nuanced understanding of meanings, contexts, and subtleties that the quantitative data could not capture on its own. By collecting and analyzing these strands in parallel, the study leveraged the strengths of both approaches while mitigating their individual limitations, ultimately producing findings that were richer, more reliable, and more contextually grounded [14,33,44].

The integration of results was carried out during the interpretation stage, where the quantitative findings were systematically compared and contrasted with the qualitative insights. Convergences were identified as points of validation, while divergences opened space for deeper theoretical reflection. This process aligns with methodological guidance in recent scholarship [32], which emphasizes the value of joint displays and integrative reasoning for strengthening validity in mixed methods research. To further enhance transparency, the research design is illustrated in a process map (Fig 1). This diagram demonstrates the parallel sequencing of quantitative and qualitative strands, their independent analysis, and the subsequent stage of integration. Such a visual representation makes the logic of the design explicit and clearly shows how the two forms of evidence were combined to address the central research question.

### 3.5. Data analysis

The study employed three complementary analytical methods: word frequency analysis, semantic network analysis, and co-word clustering to extract insights from the user comments. However, each method was selected carefully to address specific parts of the research question and perform a rigorous and systematic data analysis. These three analytical methods were integrated for their complementarity strengths. Word frequency analysis gave an idea about the data

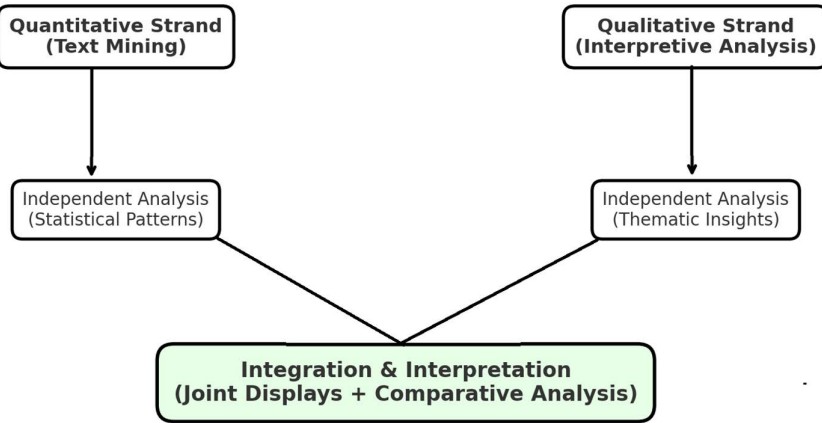

**Fig 1. Convergent Parallel Mixed Methods Design.**

breadth-wise, i.e., key terms and themes appeared. Depth was given by semantic network analysis to explore the relationships between terms, and co-word clustering provided a structured way of classifying these relationships into meaningful, motivational groups. Together, these methods are made for a thorough and well-grounded research question analysis both at breadth and depth [36,42]. The technique combines these methodologies to address the reviewers' concerns about whether inferred motivations can be derived from textual data. This multiplicity of methodological triangulation made for many layers of evidence, making the findings more reliable and interpretable [40,41]. Furthermore, the closeness of the results to the theoretical foundation of the UGT framework contributed to the robustness of the study's conclusions.

i. Word Frequency Analysis

The most often used terms in the comment corpus were subjected to word frequency analysis. This method was selected because it offers a straightforward, functional means to uncover major ideas and ideas in textual information. The themes and worries of the audience are reflected in high-frequency words, and it is helpful to begin to understand the user motivations [37,52]. A widely regarded Jieba segmentation tool was used to perform the analysis. Due to the characteristics of Mandarin that do not have spaces between words, Jieba is important as it would make it possible to segment the text into meaningful and understandable units [31]. Stop words, auxiliary terms, and low-frequency words were removed to focus the analysis on semantically significant content. Results from the word frequency analysis showed the most salient terms within the dataset and gave a foundation for subsequent methods. For instance, central themes extracted from the semantic network and co-word clustering were words that described sports performance, emotional connection, and community [53,54].

ii. Semantic Network Analysis

Semantic network analysis (SNA) was employed to examine the relationships between high-frequency words, offering insights into the conceptual structure of user comments. This method was chosen because it extends beyond simple word counts to excavate meaning between the words and the interconnected network of meaning [34,55]. A co-occurrence matrix was created to analyze, where nodes represented the words and edges represented the frequency of co-occurrence of two words within the text. The Pointwise Mutual Information (PMI) formula was used to calculate the strength of the association of word pairs to capture meaningful relationships between words instead of random co-occurrences [38,39]. The resulting semantic networks were visualized through Gephi, a network analysis software, so that nodes of central importance and thematic clusters could be identified. SNA was chosen because it could uncover

latent structure in the data, creating a richer understanding of how users cognize podcast listening experiences [56]. For example, the clusters formed for terms related to competition, camaraderie, and self-improvement, which indicate the multifaceted reasons people were likely motivated to purchase.

iii. Co-Word Clustering Analysis

High-frequency terms were subsequently grouped into thematic categories using co-word clustering to contextualize the result. The motivation for choosing this method comes from its capacity to derive empirically motivational categories from the natural organization of the data [57]. A community detection technique, the Louvain algorithm, was applied to uncover clusters of words widely related within the semantic network. The Louvain algorithm was selected because of its efficiency and effectiveness in handling large and complex datasets. By optimizing modularity, the algorithm ensured that the identified clusters were internally cohesive while remaining distinct [35,58]. Each cluster was subsequently analyzed to determine its thematic focus, which was then mapped onto the dimensions of the UGT framework. Co-word clustering was justified by its ability to move beyond descriptive analysis and better understand the relationships between motivational themes. For example, clusters related to information acquisition, emotional engagement, and social interaction emerged as key drivers of podcast consumption, aligning with the theoretical constructs of UGT [59,60].

## 4. Results

Table 1 analyzes user comments' most frequently used words, revealing dominant themes in audience discussions. The frequency analysis identified key high-usage terms within user comments, including "play" (比赛, 1,472 mentions), "win" (胜利, 1,230), and "performance" (表现, 823), indicating a dominant cognitive motivation driven by attention to competitive outcomes, athlete skill execution, and strategic insight. Terms such as "feel" (感觉, 993) and "relax" (放松, 231) highlight the emotional gratifications users derive from listening. Notably, the term "teacher" (老师, 1,221) emerged frequently, reflecting the educational tone many listeners associate with analytical commentary and professional breakdowns. Meanwhile, socially-oriented vocabulary—e.g., "family" (家人, 145) and "friends" (朋友, 138)—appeared at lower frequencies, suggesting that although podcasts serve community functions, these roles are less explicitly verbalized in comments. The semantic richness of these terms, especially when translated from Chinese, demands contextual clarification. For example, "teacher" does not merely imply instruction but often expresses deep respect for the commentator's expertise in the Chinese cultural context. This dual role—as informant and authority—distinguishes podcast hosts from casual social commentators. Clarifying these culturally embedded terms strengthens cross-linguistic interpretation and enhances the generalizability of findings. This suggests an opportunity for podcast creators to enhance social and community-building elements, fostering stronger connections among listeners. While the original word frequency and semantic network analyses reveal substantial cognitive and emotional engagement, they stop short of accounting for modern social media's subjective and symbolic power in reshaping sports discourse. This study now explicitly addresses that gap by contextualizing podcast engagement within the broader social media ecosystem. Influencers like MrBeast, who participate in high-profile

**Table 1. High-Frequency Words in Program Comments (Partial).**

| High-Frequency Words | Frequency | High-Frequency Words | Frequency | High-Frequency Words | Frequency |
|---|---|---|---|---|---|
| Play | 1472 | talk | 996 | focus | 931 |
| Say | 1275 | feel | 993 | skills | 925 |
| Win | 1230 | story | 990 | think | 897 |
| Teacher | 1221 | believe | 952 | go | 893 |
| Hear | 1021 | victory | 948 | score | 838 |

events such as the NBA All-Star Game, demonstrate the blurring of lines between entertainment creators and athletic institutions. This evolution is evident in Western sports media but increasingly relevant in China. Although Himalaya operates under stricter platform governance than YouTube or Twitch, it is not immune to this trend. Popular Chinese sports commentators increasingly blend podcasting with Weibo microblogs, Douyin clips, and live fan streams. This convergence creates multi-modal engagement where listener identity is co-constructed across platforms beyond text comments or word frequency constraints.

In contrast to Western models, where creators like Paul George and Draymond Green use personal podcasts to construct authentic, behind-the-scenes narratives, Chinese hosts often adopt a more pedagogical or structured tone. This is due to both audience expectations and media platform norms. Nonetheless, both models reveal how subjective storytelling, not just objective match analysis, is central to engagement. These nuanced dynamics—where emotional resonance, influencer credibility, and fan co-creation converge—highlight that sports podcasts serve as content and cultural touchpoints. Therefore, the paper extends beyond computational patterns to address symbolic interactionism in digital sports media: how meaning is shaped, shared, and validated across interconnected platforms. This layered interpretation significantly advances the understanding of sports communication in a digital-first environment, aligning Chinese trends with global shifts while acknowledging their unique characteristics.

Fig 2 visualizes the interplay between podcast consumption's cognitive, emotional, and social dimensions. The networks reveal overlapping motivations, where personal aspirations for self-efficacy intersect with communal experiences of social interaction and emotional resonance. Terms such as victory and achievement, which sound good when used, match perfectly with group identity and enthusiasm for sports. Commensurate with the density of connections in these networks, audience motivations for these podcasts are socially and emotionally integrated, emphasizing the human reason to value podcasts as not just singular, stand-alone listening experiences but as such interconnections in listeners' lives. Meanwhile, the visualizations themselves do not create actionable insights for content creators. The usefulness of these patterns could be significantly increased if they are translated into practical recommendations such as optimal content lengths or thematic focuses. While the visualized networks are effective at coding the richness of motivational dynamics, they do a poor job of fleshing out their implications for content strategy. It will be interesting to see how the contents of these insights can shape podcast design, such as producing content at scale for a particular audience or optimizing the formats of the episodes to increase engagement.

Table 2 provides view and listener counts, episodes, plays, and comments for five of the Himalaya platform's most subscribed sports podcast channels. Such points out the essence of the difference in terms of audience coverage and channel activity trends. *Yang Yi Radio* stands out as the station that garners the most plays, with 110 million despite the few subscribers or 21,600. Such a substantial divergence would indicate that the channel can reach an even broader audience than the current subscribers, implying that the channel was promoted, engaging content was promoted, and word-of-mouth communication was established. Also, the 11,000 comments it elicits indicate a very social audience, which indicates a high audience interaction per play. Still, the difference between the play number and the number of subscribers indicates untapped potential in getting the play frequency to translate to subscriber frequency, which can be done by offering subscriber-only content or raising subscriber perks.

Conversely, the *Sports Basketball Show* recorded the highest number of subscribers at 80,000, but PED only 54 9 million plays and considerably low interaction with a comment rating 2256. This suggests possible difficulties in converting subscribers into active listeners and engaging them. This disparity might be attributed to content that does not meet the listeners' expectations or little advertising. Solving such problems could increase the usage of the subscriber base effectively and might lead to increased play counts and interaction.

Other related channels, including *Nan's Talk* and *CBA Fans Radio*, show moderate activity. Nan's Talk has 825 episodes, representing the highest number across all the presented channels, reaching 100 million plays. This has significant implications, including arguing that ISG has a vast content program that keeps many audience members hooked.

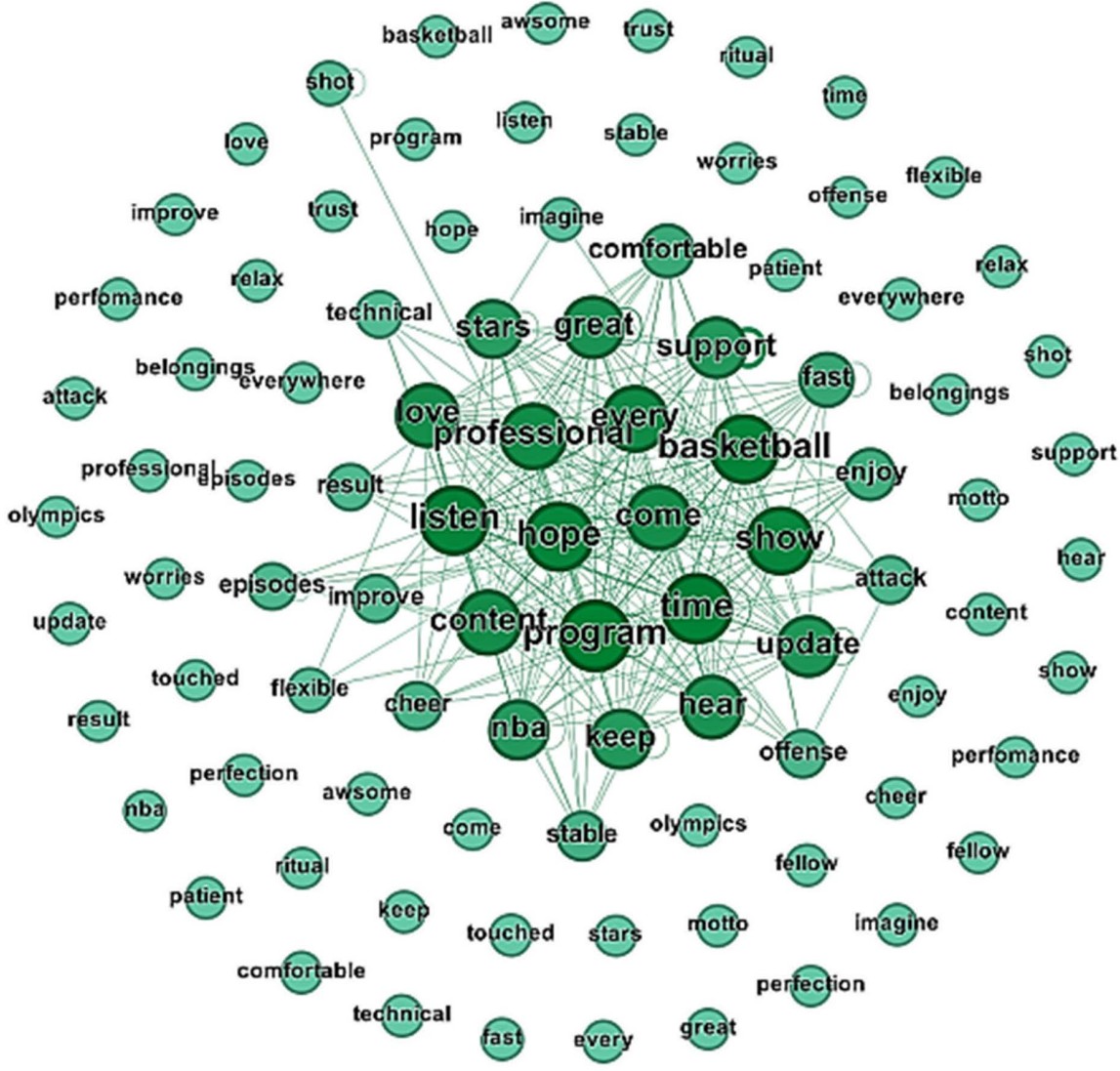

**Fig 2. Thematic Network of Emotional Embodiment of Athletic Competitiveness and Participation.**

However, the low comment volume of 1,450 shows a low activity level and many potential consumers and viewers, suggesting more engaging content formats. Similarly, *CBA Fans Radio* achieves 78.9 million plays with 23,000 subscribers and 448 episodes yet generates only 852 comments, pointing to limitations in its ability to cultivate interactive communities. Finally, *Football First Perspective*, with 47,000 subscribers and 36 million plays, demonstrates a balanced performance but has opportunities to expand its reach and deepen engagement by integrating interactive features and social media promotion. The most significant gap in this analysis is the disparity between subscriber counts, play counts, and comment volumes. Channels with high play counts, like *Yang Yi Radio*, may lack strategies to convert passive listeners into active subscribers, while others, such as *Sports Basketball Show*, struggle to activate their large subscriber bases. This points to a need for tailored content and marketing strategies to enhance audience engagement and foster loyalty.

**Table 2. Channel Name and Comment Data.**

| Channel Name | Subscribers | Episodes | Plays(million) | Comments |
|---|---|---|---|---|
| Sports Basketball Show (体坛篮球秀) | 80,000 | 533 | 54.9 | 2,256 |
| Yang Yi Radio (杨毅电台) | 21,600 | 372 | 110 | 11,000 |
| Football First Perspective (足球第一视角) | 47,000 | 529 | 36 | 1,701 |
| Nan's Talk ([无用学]很"男"说) | 24,000 | 825 | 100 | 1,450 |
| CBA Fans Radio (开场shao CBA球迷电台) | 23,000 | 448 | 78.9 | 852 |

Fig 3 illustrates the semantic network of user comments, showing the relationships between frequently used terms. Central nodes like "game," "win," and "team" dominate the network, reflecting a shared focus on core elements of sports events. The network density of 0.234 suggests a moderately interconnected structure, indicating thematic diversity while maintaining centrality around key topics. Emotionally charged words, clusters of words dealing with emotion like 'hope,' 'heroes,' and 'moved,' tell us how emotionally charged listening engagement is. Such terms imply that sports podcasts speak to the ears of their respective audiences through tales of triumph, struggle, and motivation. Moreover, the appearance of terms like 'coach' and 'group' highlights the social aspect, in which audiences conceptualize collective sports experiences and agree on the feeling of being members of the group. However, it does not provide any sense of the granularity of how these thematic clusters differ by demographics or genre. For example, audiences of younger age may want to emphasize some of the emotional themes, whereas audiences of older age would wish to emphasize more of the informational aspects. Such variations could be further understood by future research that segments the data by listener profiles. The semantic network analysis successfully captures thematic richness in user comments but does not capture listener preference diversities or behavioral variations. Some insight into these dynamics could be gained by incorporating other data sources, like user surveys or demographic segmentation.

Table 3 categorizes user motivations into five primary dimensions: Information Acquisition, Entertainment and Leisure, Social Interaction, Self-efficacy, and Media Convenience. The top category above is information acquisition, which incorporates 7,700 mentions focused on real-time updates, game analysis, and athlete insight. This tells us why sports podcasts help audiences to obtain all the information they need about sports. The most important business aspect is that 5,957 mentions focused on Entertainment and Leisure, clearly showing how recreational podcast listening can be. "Relax," "enjoy," and even "tradition" and "spirit" are terms that relate to the audience's desire for the escape and emotional relief that comes with podcasts. Writing about 10,271 mentions, Social Interaction draws attention to the communal elements of podcast listening as listeners interact with podcasters and sports celebrities and form group modes of identity. Self-efficacy, the largest category with 16,803 mentions, emphasizes personal growth and inspiration as listeners seek to emulate athletic skills and derive confidence from sports narratives. Ultimately, the vast number of mentions in Media Convenience (8,054) hints at the sheer practical benefits of podcasts, from accessibility and flexibility that ease the ability to weave content into the daily schedules. The categorization reveals the polymeric motivations for listeners but not how these dimensions mutually come into play. Others fail to examine the relationship between media convenience and emotional engagement. Understanding such intersections may give greater clarity about how podcasts meet the challenging needs of listeners.

## 5. Discussion

This study found that self-efficacy is the most dominant user motivation driving Chinese sports podcast engagement, followed by social interaction and media convenience. This is consistent with Perks et al. (2019), who observed that sports

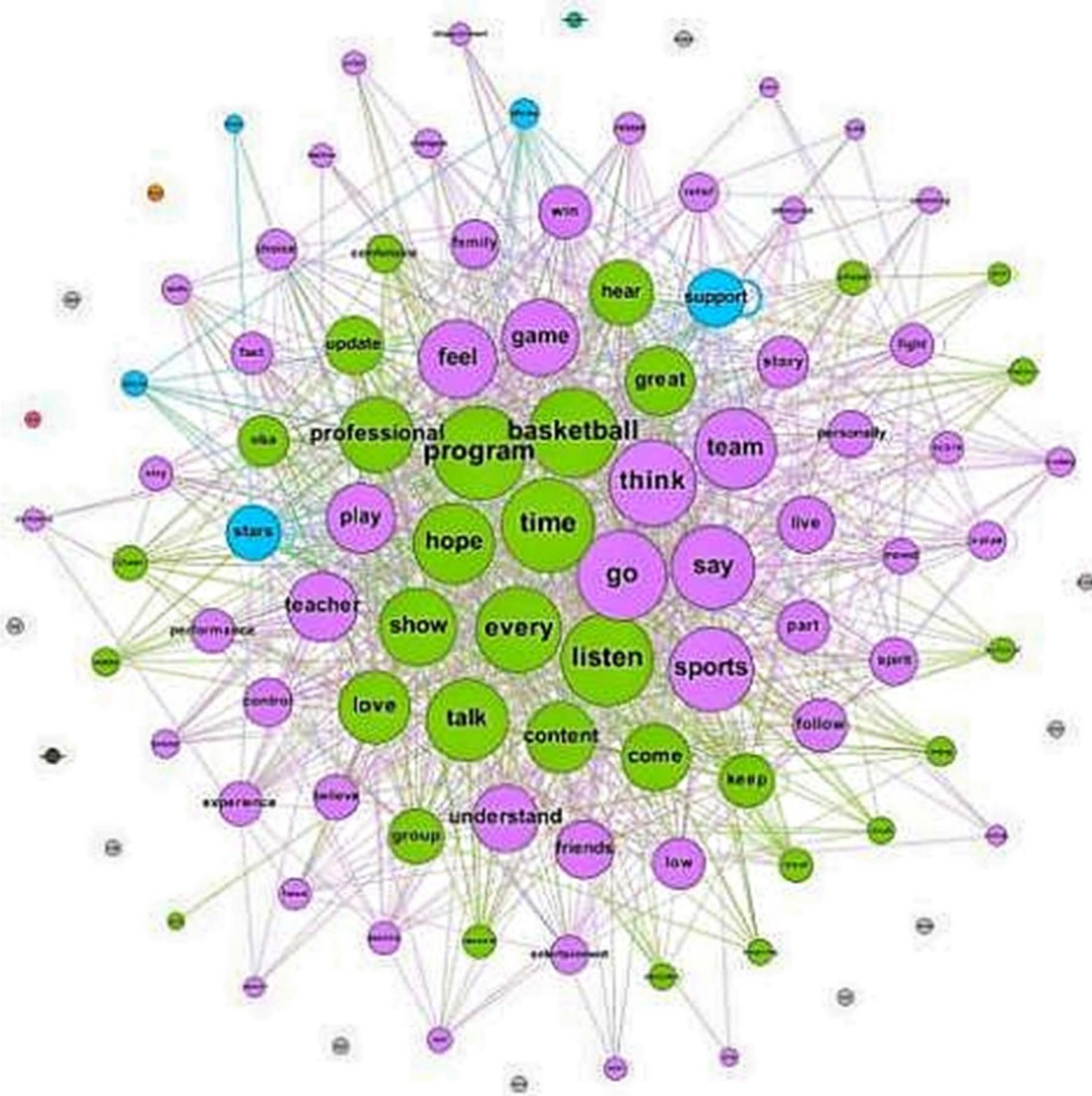

**Fig 3.  The Overall Semantic Network of Sports Podcast Program Reviews.**

fans often seek psychological reinforcement and identity validation through mediated sports experiences. However, the prominence of the term 'teacher' (老师) in Chinese user comments introduces a culturally specific layer absent from prior U.S.-based research. Unlike the casual 'host' model in Western podcasts, Chinese audiences confer status and epistemic authority to podcast presenters, positioning them as trusted mentors—reflecting Confucian dynamics in knowledge consumption. Additional comparisons now include works by Chan-Olmsted & Wang (2022), Tendinha et al. (2021), and Koob (2021), examining how this study's findings align with, diverge from, or extend previous models of podcast engagement and media gratification. Each motivational theme is now debated about theoretical constructs (e.g., UGT, Social Identity Theory) and empirical analogs, adding critical depth. For example, the discussion on emotional resonance is connected to sports narrative theory, and the analysis of group identity is cross-referenced with Trepte and Loy's work on collective fandom behavior in digital communities.

**Table 3. Categorization Results of Sports Podcast Audience Comment Content.**

| Primary Category (Frequency) | Secondary Category (Frenquency) | High-frequency Words (Frenquency) |
|---|---|---|
| **Information Acquisition (7700)** | Game status (1,810) | result(674),shot(463),live(255),score(236),moment(120),conflict (62) |
| | Athlete information (1,863) | team(462),experience(457),fast(414),record(288),scorer(134),injury(86),transfer (22) |
| | Sports knowledge and common sense (2,654) | teacher (1,221),basketball(401),tennis(289),swimming(203),olympics(189),NBA(118),football(117), CBA (64),rules (52) |
| | Commentary and analysis (1,373) | perfomance(312),attack(213),technical(192),offense(170),comment(105),break(103),technique(88),analysis(76),practical (58),foul (56) |
| **Entertainment and Leisure (5957)** | Leisure Activity (1,049) | game(423),enjoy(188),entertainment(165),pleasant(129),choice(83),different (61) |
| | Emotional Regulation (1,281) | awsome(272),relax(231),disappoinment(214),tension(199),control(173),undecided(123),mindset 69 |
| | Emotional Relaxation (1,399) | hope(334),relaxed(189),relief(185),worries(177),comfortable(173),breathe(113),calmer(93),Ease(79),effortlessly 56. |
| | Cultural Value (2,228) | Spirit(732),value(529),tradition(412),ritual(334),motto(221) |
| **Social Interaction (10271)** | Interaction with podcaster (3,147) | Yang(757)t,update(535),program(505),show(347), support(258),host(229),professional(220),content(187), view(109) |
| | Adoration of sports celebrities (1,328) | cool(270),champion(230),stars(145),symbol(127),idol(108),fans(102),charismatic(88),coach(77),MaLong (73),MVP (60),Gu (48) |
| | Group Identity (2,315) | family(727),friends(483),group(432),fellow(322),Belongings(175),loyal(89),objective(87) |
| | Emotional Resonance (3,481) | feel(993),touched(762),heroes(439),moved(441),understand(320), inspiring(289),imagine(237) |
| **Self-Efficacy (16803)** | Mastery Experiences (2,774) | perfomance(823),confidence(596),stable(521),perfection(379),skills(283),athleticism(172) |
| | Vicarious Experiences (5,355) | play (1,472),win(1230),victory(948),score(838),fight(369),part(237),low(179),mirror(82) |
| | Verbal Persuasion (6,203) | believe(952),focus(931),go(893),follow(833),keep(724),trust(676),come(425),ahead(337),stay(243),invest(189) |
| | Self-Improvement (2,471) | skills(925),improve(572),strategy(521),patient(318),endurance [13] |
| **Media Convenience (8,054)** | Easy-accessibility and companionship (2,894) | hear (1,021),story(990), everywhere(486), flexible(245), visit(152) |
| | Platform to express (5,160) | say (1,275),talk(996),think(897),personally(727),episode(523),cheer(431),unfair(312) |

The findings of this study provide a detailed exploration of user motivations for sports podcast consumption on the Himalaya platform, contextualized within the framework of UGT. These findings were complemented by the responses from the user comments, which provide nuanced insights into how podcasts fulfill listeners' informational, emotional, and social needs. Analyzing the results alongside these user sentiments shows that sports podcasts serve as personal and communal spaces, fostering cognitive engagement, emotional resonance, and social belonging. Information acquisition emerged as a primary driver for podcast consumption, with users seeking detailed analyses, game updates, and expert commentary. This aligns with one listener's response:

*"Every time I tune in, it feels like I'm courtside at an NBA game. The hosts break down all the strategies and game details like real pros."*

Such feedback underscores sports podcasts' role as immersive platforms replicating the depth and excitement of live sports events. Previous studies, such as those by Rime et al. [19], highlighted podcasts' informational appeal, particularly in providing specialized insights unavailable through traditional media. Podcasts, as another listener pointed out, can dissect "the smallest moves" and go down to experience "team culture," among other demands from sports fans that podcasts are ideally situated to satisfy concerning in-depth expertise. However, there is a gap in how these podcasts appeal to casual fans versus avid enthusiasts [19,20]. While listeners appreciate detailed breakdowns, some noted that the analysis remains accessible even to non-experts:

*"The analysis is so on point that even a casual fan like me can totally follow along."*

This dual appeal reflects the balance sports podcasts strike between sophistication and inclusivity, a feature that distinguishes them from traditional broadcasts. Such accessibility is necessary to broaden the audience base, matching Koob [61] assumption that media content needs to adapt suitably to users' different needs. In addition to cognitive engagement, sports podcasts are a significant means for an audience to vent emotions. As one respondent put it:

*"This podcast is like fuel for my soul! Every time I listen, it's like getting a serious energy boost."*

This attitude fits the results of Table 2, where emotionally charged words like "hope," "relaxed," and "confidence" showed up often. Podcasts' capacity to inspire and uplift, mainly through tales of athletes' tenacity and successes, reflects their involvement in satisfying emotional needs. This is further echoed by another listener who remarked:

*"Every time Yang talks about an athlete's preparation story, I feel like I'm right there on the track, pushing alongside them."*

Such stories show how podcasts go beyond simple entertainment to become vehicles for emotional emancipation, therefore fostering a feeling of personal connection and motivation. This research supports Tendinha et al. [27], who indicated that sports narrative motivates humans. However, this research further deepens this knowledge by looking at the temporal link between emotional involvement and self-efficacy. One listener said:

*"This show is like my 'confidence booster,' making me feel stronger and more self-assured when I'm up against a challenge."*

This emphasizes sports podcasts' unique role in improving listeners' resilience and self-perception—dimensions sometimes disregarded in current research. Although podcasts for schooling have been a tool for boosting self-efficacy, this

study reveals that sports podcasts can also enable one to progress and keep on for oneself. Social contact was another prevailing topic related to sports consumption as community consumption [21,22]. User comments frequently referenced feelings of camaraderie and belonging, as illustrated by one listener's remark:

*"It feels like I'm right there with a group of super passionate sports fans, fighting side by side."*

This corresponds with Social Identity Theory (SIT), which holds that group affiliations strengthen social ties and collective identity [30,62]. The ability of sports podcasts to foster such connections is particularly evident in comments like:

*"Being part of this sports community makes me feel so important, like I'm actually at the game, cheering for my team."*

These feelings highlight the transforming power of podcasts as online venues for common experiences, as listeners go beyond geographical distances to create emotional bonds with other enthusiasts. The Himalaya platform's interactive elements, such as comment sections and live question and answer events, accentuate this community element even more. As one listener said:

*"Every time I finish listening, I feel like I understand the game a little better. When I watch the match with my friends, I can throw in a few more comments."*

This emphasizes that podcasts help share knowledge and enhance social contacts, increasing listeners' relationships. Chaves-Yuste and de-la Peña [57] and Perks et al. [58] noted that podcasts act as accelerators for group conversations and group learning and also identified similar results. However, the current study adds depth by exploring how these interactions enhance group dynamics, as noted by one respondent:

*"And every time I hang out with my buddies to talk basketball, we have more topics to discuss and our bond feels even stronger."*

This study brings a key insight: the unique trajectory of sports Internet communication in China compared to Euro-American counterparts. While U.S. and European podcasts often rely on decentralized content ecosystems (e.g., Patreon support, independent creators), Chinese sports podcasts thrive within centralized, super-app ecosystems like Himalaya, which combine audio streaming, social interaction, and e-commerce. This integration shapes communication styles and content delivery. Chinese platforms emphasize real-time interaction (e.g., bullet comments, live Q&A), short-form social discourse, and indirect engagement through hierarchical learning (e.g., podcast hosts as experts/teachers), while Western podcasts emphasize longer narrative arcs and dialogic engagement between hosts and audiences. Furthermore, in China, audience discussions are shaped by collectivist norms, often reflecting shared group sentiments over personal anecdotes. This contrasts with Western podcast reviews, where individual voice and identity often dominate. These differences indicate that while the motivations (as described by UGT) may be structurally similar, the communicative texture and social affordances are culturally contingent. The lack of peer-to-peer interaction in some Chinese podcasts may limit horizontal community formation compared to Reddit or Discord-driven fandoms in the West. To deepen this comparative understanding, future research should examine cross-platform user behavior, especially between global and Chinese podcasting platforms, including Spotify, Apple Podcasts, and Himalaya, to identify convergence or divergence in social communication formats. Recognizing these differences enriches platform strategy and theoretical framing, emphasizing that socio-technical context always mediates media behavior. Dealing with this disparity could help one to have a more complete awareness of the part sports podcasts play in forming digital and physical fan bases. Another important discovery is media convenience, which underlines much of podcast use. Table 3 clarifies that incorporating

podcasts into listeners' daily activities depends critically on the accessibility and adaptability of the Himalaya platform. One user succinctly captured this sentiment:

*"This show is like my safe haven! Every time I listen, all the little daily annoyances and work stress just fade away."*

This shows how well the platform can deliver flawless, on-demand content fit for consumers in a limited time. Emphasizing their attraction to multitasking viewers, Yang and Tasi [29] similarly underlined the flexibility of podcasts as a primary driver of their success. The present research points out a possible discrepancy in how convenience interacts with other motivating factors. Although listeners like the simplicity of access, it is not apparent how this element affects more profound forms of involvement, such as social interaction or emotional connection [24]. Investigating these crossroads might expose fresh approaches to improve customer satisfaction and retention. The unique mix of these motivating elements, such as cognitive, emotional, social, and pragmatic, positions sports podcasts as a diverse means of audience involvement. This is particularly evident in user comments that integrate multiple gratifications, such as:

*"It just fires me up, like I'm getting stronger along with it. This isn't just about listening to a show—it's an emotional outlet and a place I belong, where I feel the power and warmth of sports."*

This holistic experience reflects sports podcasts' evolving role as platforms catering to diverse and overlapping user needs. The findings also underscore the importance of content diversity in sustaining audience interest. As one listener observed:

*"This channel is packed with all kinds of content—from international tournaments to street ball, it's got it all!"*

Different formats, such as fan calls and expert interviews, ensure podcasts remain interesting and dynamic. This aligns with Menon [63] perspective that media material should constantly shift to captivate viewers. Still, Table 1 demonstrates that the relatively modest engagement rates for different channels indicate that not all podcasts use this diversity well. Ultimately, depending on its results and enhanced by user feedback, the study emphasizes the several elements impacting sports podcast consumption. This study thoroughly examines how sports podcasts meet cognitive, emotional, social, and practical demands, contextualizing these findings within accepted theories and contrasting them with existing literature. However, there are significant gaps, particularly in studying demographic differences, offline activities, and the interconnections of driving factors. Filling these gaps is critical for advancing research on digital media consumption and optimizing podcast content for varied audiences.

This study extends the UGT by integrating it into a high-context, platform-dominated media culture like China's. It introduces culturally grounded motivational categories (e.g., 'teacher trust') and demonstrates how self-efficacy operates not just as personal confidence, but as socially aspirational learning tied to public figures in the podcasting space. However, this study makes several noteworthy contributions to academic and practical discourse. Theoretically, it extends the application of the UGT to the context of sports podcasts in China, providing insights into how digital platforms cater to diverse audience needs. Methodologically, the study integrates advanced text analysis techniques with user sentiment analysis, offering a robust approach to understanding audience behaviors.

For content creators and platforms, the findings suggest that emotionally empowering, informatively rich, and socially participatory content is key to sustaining user engagement. Features like live Q&As, fan discussion threads, and story-driven analysis can enhance both retention and community cohesion. For marketers, segmenting content based on motivational clusters—information-seekers vs. identity-builders—can yield more targeted promotional strategies."

Practically, the findings provide actionable recommendations for podcast creators and platform developers, highlighting strategies to optimize content, foster engagement, and expand reach. People working in content creation, platform

development, and media analysis will find this study's findings helpful. Content creators must provide content that combines professional insights with compelling stories since people are looking for informational and emotional rewards. By exhibiting a variety of content formats, including live Q&A sessions, athlete interviews, and fan call-ins, platforms like Himalaya may utilize this information to match the different interests of their audience. Additionally, the chance to build more substantial fan groups through interactive features and offline meetups is highlighted by the social aspects of podcast listening. In order to uncover complex audience behaviors, the study stresses the importance of combining theoretical frameworks with advanced analytical tools. This study provides a strategy for studying digital media consumption in different circumstances by mixing qualitative and quantitative methodologies.

Several suggestions for filling the gaps have emerged; content providers should work to increase the number of interactive and dynamic episodes available to both casual listeners and die-hard fans. Platforms should look for ways to customize content delivery through advanced algorithms to ensure that customers are matched with episodes that match their preferences. Supporting in-person events, such as fan gatherings or live-streamed performances, can bring listeners closer. Future research using longitudinal studies may highlight the changing nature of motives and participation to a greater extent. Expanding the scope of the study to include different platforms and ethnic backgrounds would make the results more applicable. More specifically, targeted content tactics may be possible with demographic segmentation, which may also show disparities in audience motives.

## 6. Conclusion

This study demonstrates the robustness of the UGT in explaining Chinese sports podcast engagement, while also extending its applicability beyond Western contexts. Drawing on over 13,000 user comments, the analysis revealed clear cognitive, emotional, and social gratifications embedded in audience discourse. Word frequency, semantic networks, and co-word clustering consistently mapped onto UGT categories, confirming that motivations such as information acquisition, emotional resonance, and social interaction are not only present but actively articulated by listeners.

Notably, self-efficacy emerged as the dominant motivational theme, highlighting how podcasts serve as aspirational tools that foster confidence, identity construction, and personal growth. Cultural specificity was equally significant: the recurrent motif of "teacher" underscored the importance of authority, respect, and hierarchical knowledge transmission in shaping engagement patterns within a high-context culture. By integrating computational text mining with qualitative interpretation, this study enhances UGT's explanatory power, demonstrating that digital audiences utilise podcasts for both entertainment and the construction of agentic, identity-driven narratives. The findings underscore the role of podcasts as accessible, emotionally engaging media that strengthen habitual engagement and loyalty. Theoretically, the study enhances UGT's cross-cultural validity; practically, it provides guidance for podcast creators seeking to design content that strikes a balance between cognitive depth and emotional and social resonance.

Future studies should explore how listener motivations evolve over time using panel data, and include comparative cross-platform analysis (e.g., Himalaya vs. Spotify) to assess global vs. local engagement patterns. Including gender, age, and geographic segmentation will also enrich understanding. Future research can improve the podcasting experience for all types of listeners by addressing the identified restrictions and implementing the proposed recommendations, allowing us to delve deeper into the complexities of digital media consumption. A more complete understanding of podcast listening might be obtained by investigating the interplay between different motivating elements, such as social and emotional incentives.

## Supporting information

**S1 Table. Alignment of Research Questions, Methods, and Results.** S1 Table presents the alignment between the study's research questions, methodological approaches, and key findings, showing how cognitive, emotional, and social gratifications were identified and linked to audience engagement on Chinese sports podcasts.
(DOCX)

## Author contributions

**Writing – original draft:** Fangni Li.

**Writing – review & editing:** Fangni Li.

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
