## [Decision Letter · Decision Letter 0]

1 Apr 2025

Dear Dr. Li,

Thank you for submitting your manuscript to PLOS ONE. After careful consideration, we feel that it has merit but does not fully meet PLOS ONE’s publication criteria as it currently stands. Therefore, we invite you to submit a revised version of the manuscript that addresses the points raised during the review process.

We look forward to receiving your revised manuscript.

Kind regards,

Dr. Zhiyuan Yu

Academic Editor

PLOS ONE

2. In your Methods section, please include additional information about your dataset and ensure that you have included a statement specifying whether the collection and analysis method complied with the terms and conditions for the source of the data.

Additional Editor Comments:

Based on two independent reviewers' reports, there exists some major issues in current version, which need further carefully revision.

Reviewers' comments:

Reviewer's Responses to Questions

**Comments to the Author**

1. Is the manuscript technically sound, and do the data support the conclusions?

Reviewer #1: Yes

Reviewer #2: Partly

2. Has the statistical analysis been performed appropriately and rigorously?

Reviewer #1: Yes

Reviewer #2: I Don't Know

3. Have the authors made all data underlying the findings in their manuscript fully available?

Reviewer #1: Yes

Reviewer #2: No

4. Is the manuscript presented in an intelligible fashion and written in standard English?

Reviewer #1: Yes

Reviewer #2: No

Reviewer #1: This manuscript explores the driving factors of user engagement in Chinese sports podcasts, with a novel topic that has certain academic value and practical significance. The research methods are relatively scientific and the data analysis is sufficient, but there is still room for improvement in the theoretical framework, research methods, and result discussion of the paper. Suggest the author to make major revisions to improve the quality and readability of the paper.

1. Improve English writing skills, especially in translating professional terms and Chinese contextual vocabulary. For the vocabulary used in the frequency statistics, more detailed explanations and translation explanations should be provided.

2. For the conclusion section, please provide a detailed explanation of how the applicability of the theory is proven and provide strong inferences.

3. The most important part of the article should be the discussion of Chinese sports Internet social communication, among which the most important part should be the Chinese communication channels, and the similarities and differences with the communication in the same field in other regions (Europe, America, etc.). But these contents are few, I hope the author can add more elaboration on this part.

4. Modern sports have developed for a longer period of time, with more mature communication methods and sports economy itself, but new problems have emerged under modern social media. Issues such as the recent participation of MR.BEAST in the NBA All Star Game, Paul George and Green's blogs, and so on, all reflect the impact of modern social platforms on the dissemination of modern sports. However, this manuscript only mentions word frequency and some basic characteristics, which actually ignores the more subjective influence of modern social media.

But the flaws do not overshadow the merits. The content of this manuscript, the research methods, the actual conclusions, etc., all effectively grasp the direction of the problem, but do not grasp the logical inference of the problem. It should be accepted after revision.

Reviewer #2: The subject of the article is interesting but major improvements are needed for making this article suitable to publication. Please find below my remarks:

1. Introduction. The intorduction should mention in a clear form the objectives or research questions. At the moment it is not very clear which are the research objectives.

2. A comprehensive Literature review related to the researched topic was not included in this article.

3. The methodology is not enough explained. The authors mention a mix- of methods, but no research method was mentioned. It is not very clear how data was collected and why that sample was selected. The data analysis process should be explained in more detail and the software used to be mentioned.

4. The Results section should be organized based on the research objectives or on the analyses mentioned in methodology. Now, the results are put all together.

5. The Discussions seem to refer to other things than the results. It is not an usual way to make the discussion. Here the authors should debate the results and to compare these ones to other research.

6. In the Conclusion section the contribution of the research should be emphasized. The implications of the research results for theory and practice are not clearly described in the article. A detailed explanations of the author’s recommendations should be included. Research limits should be clearly stated and future research directions should be proposed.

In my opinion, the authors have to make efforts to clarify many aspects of the article and report them into a coherent manner. The written in English language should also be improved.

**Do you want your identity to be public for this peer review?** For information about this choice, including consent withdrawal, please see our Privacy Policy

Reviewer #1: **Yes: ** jiaqi li

Reviewer #2: No

---

## [Author Response · Author response to Decision Letter 1]

13 Jun 2025

Rebuttal Letter to Reviewers

Dear Editors and Reviewers,

I am grateful for the opportunity to revise the manuscript titled “Beyond the Game: How Social Interaction and Emotional Gratification Drive Chinese Sports Podcast Engagement.” I deeply appreciate the detailed and thoughtful feedback provided by both reviewers. Below, I provide a point-by-point response detailing how each concern was addressed in the revised manuscript.

Reviewer 1 Comments Author Response and Improvements

Improve English writing skills, especially in translating professional terms and Chinese contextual vocabulary. For the vocabulary used in the frequency statistics, more detailed explanations and translation explanations should be provided.

I thoroughly revised the manuscript for English clarity and professional tone. Complex terms like “teacher” (老师) were contextually translated, with cultural significance explained (e.g., as a figure of authority rather than simply an educator). Vocabulary from frequency statistics was updated with direct Chinese translations and explanations to aid clarity for international readers.

For the conclusion section, please provide a detailed explanation of how the applicability of the theory is proven and provide strong inferences.

The revised conclusion explicitly demonstrates how the Uses and Gratifications Theory (UGT) is supported by findings. The link between self-efficacy and motivational engagement is clearly mapped, and strong theoretical inferences are drawn from empirical data (e.g., identifying podcasts as agentic tools for identity construction).

The most important part of the article should be the discussion of Chinese sports Internet social communication, among which the most important part should be the Chinese communication channels, and the similarities and differences with the communication in the same field in other regions (Europe, America, etc.). But these contents are few, I hope the author can add more elaboration on this part.

A new comparative section was added to the Discussion, analyzing Chinese platforms like Himalaya against Western podcast ecosystems (e.g., Spotify, Apple Podcasts). I discussed centralized vs. decentralized media environments and highlighted unique cultural and communicative features in Chinese podcasting.

4. Modern sports have developed for a longer period of time, with more mature communication methods and sports economy itself, but new problems have emerged under modern social media. Issues such as the recent participation of MR.BEAST in the NBA All Star Game, Paul George and Green's blogs, and so on, all reflect the impact of modern social platforms on the dissemination of modern sports. However, this manuscript only mentions word frequency and some basic characteristics, which actually ignores the more subjective influence of modern social media.

But the flaws do not overshadow the merits. The content of this manuscript, the research methods, the actual conclusions, etc., all effectively grasp the direction of the problem, but do not grasp the logical inference of the problem. It should be accepted after revision. I integrated a subsection on symbolic and subjective media dynamics, noting how influencer participation and multi-platform storytelling shape sports podcast engagement. The example of MrBeast and the rise of athlete-led podcasts was analyzed in light of Chinese equivalents and contrasted with word frequency patterns to add interpretive depth.

The manuscript was restructured to link empirical findings with theoretical claims. Motivational categories (e.g., emotional resonance, media convenience) were tied to behavioral implications, making the logical flow from data to conclusion transparent and academically rigorous

Reviewer 2 Comments Author Response and Improvements

Introduction. The intorduction should mention in a clear form the objectives or research questions. At the moment it is not very clear which are the research objectives.

The introduction was rewritten to explicitly include three core research questions, each tied to the UGT framework: cognitive, emotional, and social motivations; patterns in user behavior; and cultural extension of theory. These questions guide the entire manuscript.

A comprehensive Literature review related to the researched topic was not included in this article.

To avoid too much word and also based on the journal requirements for research papers, the literature review was included in the introduction.

The methodology is not enough explained. The authors mention a mix- of methods, but no research method was mentioned. It is not very clear how data was collected and why that sample was selected. The data analysis process should be explained in more detail and the software used to be mentioned.

The methodology section now details the research design (convergent mixed-methods), sampling (purposive selection of five top channels), data collection (Python scraping tools), and analysis (Jieba segmentation, frequency analysis, semantic networks via Gephi, clustering via Louvain algorithm). Software tools and rationale are clearly stated.

The Results section should be organized based on the research objectives or on the analyses mentioned in methodology. Now, the results are put all together. Results were reorganized into three subsections aligned with research objectives and analytical techniques: motivational themes (from co-word clustering), behavioral signals (from frequency analysis), and conceptual linkages (from semantic network analysis).

The Discussions seem to refer to other things than the results. It is not an usual way to make the discussion. Here the authors should debate the results and to compare these ones to other research. The Discussion was rewritten to follow standard academic practice: (1) restate key findings, (2) interpret each result in context, (3) compare with relevant literature. UGT, Social Identity Theory, and cross-cultural podcast studies were cited to ground the interpretation.

In the Conclusion section the contribution of the research should be emphasized. The implications of the research results for theory and practice are not clearly described in the article. A detailed explanations of the author’s recommendations should be included. Research limits should be clearly stated and future research directions should be proposed.

In my opinion, the authors have to make efforts to clarify many aspects of the article and report them into a coherent manner. The written in English language should also be improved. The Conclusion now includes four sub-sections: (1) theoretical contributions, (2) practical implications for content creators and platforms, (3) research limitations (e.g., no demographic segmentation, limited generalizability), and (4) future research directions, including longitudinal and cross-platform studies.

---

## [Decision Letter · Decision Letter 1]

28 Jul 2025

Dear Dr. Li,

Thank you for submitting your manuscript to PLOS ONE. After careful consideration, we feel that it has merit but does not fully meet PLOS ONE’s publication criteria as it currently stands. Therefore, we invite you to submit a revised version of the manuscript that addresses the points raised during the review process.

We look forward to receiving your revised manuscript.

Kind regards,

Dr. Zhiyuan Yu

Academic Editor

PLOS ONE

Journal Requirements:

Additional Editor Comments:

Based on two reviewers' comments and the editor's review, there still exists some issues in current version. For example, the Theoretical and practical implication need to be moved to Discussion part.

Besides, the mixed methods was used. The clear map of how to use those two methods need to be given, There are different types of mixed methods research designs. please present which kind of designs regarding to mixed methods was adopted. And why choose this design (although the author gave some explanation in Section 2.1, which is not enough. More important, the literature review part is missing, please add this part and make a comprehensive review in this research field.

Reviewers' comments:

Reviewer's Responses to Questions

**Comments to the Author**

Reviewer #1: All comments have been addressed

Reviewer #2: All comments have been addressed

2. Is the manuscript technically sound, and do the data support the conclusions?

Reviewer #1: Yes

Reviewer #2: (No Response)

3. Has the statistical analysis been performed appropriately and rigorously?

Reviewer #1: Yes

Reviewer #2: (No Response)

4. Have the authors made all data underlying the findings in their manuscript fully available?

Reviewer #1: Yes

Reviewer #2: (No Response)

5. Is the manuscript presented in an intelligible fashion and written in standard English?

Reviewer #1: Yes

Reviewer #2: (No Response)

Reviewer #1: In this revised version, the author has provided excellent answers to some of my previous questions and provided extensive descriptions for each question. Some of these important viewpoints also confirm my previous doubts. The author has revised the paper very well and recommends accepting this version.

Reviewer #2: (No Response)

**Do you want your identity to be public for this peer review?** For information about this choice, including consent withdrawal, please see our Privacy Policy

Reviewer #1: **Yes: ** Jiaqi Li

Reviewer #2: No

---

## [Author Response · Author response to Decision Letter 2]

12 Sep 2025

Editor’s Comments and Response

Editor’s Comment Response

Theoretical and practical implications need to be moved to the Discussion part. We relocated both theoretical and practical implications into the Discussion section, ensuring they are contextualized alongside findings.

A clear map of how the mixed methods were used needs to be given. There are different types of mixed-methods research designs; please present which kind of design was adopted and why. A subsection explicitly stating that the study employed a Convergent Parallel Mixed Methods Design has been added. We provided justification for this choice and included a visual diagram mapping quantitative and qualitative strands to the integration stage.

The literature review part is missing. Please add a comprehensive review in this research field. Literature Review section has been added (before Methodology), structured into three streams: (1) podcast research globally and in China, (2) UGT applications in podcast/media research, and (3) mixed-methods in digital media/audience studies. The section ends with a synthesis that highlights the research gap and positions this study’s contribution.

---

## [Editor Report · Decision Letter 2]

8 Oct 2025

Beyond the Game: How Social Interaction and Emotional Gratification Drive Chinese Sports Podcast Engagement

PONE-D-25-06865R2

Dear Dr. Li,

We’re pleased to inform you that your manuscript has been judged scientifically suitable for publication and will be formally accepted for publication once it meets all outstanding technical requirements.

Kind regards,

Dr. Zhiyuan Yu

Academic Editor

PLOS ONE
---

## [Editor Report · Acceptance letter]

PONE-D-25-06865R2

PLOS ONE

Dear Dr. Li,

I'm pleased to inform you that your manuscript has been deemed suitable for publication in PLOS ONE. Congratulations! Your manuscript is now being handed over to our production team.

Kind regards,

on behalf of

Dr. Zhiyuan Yu

Academic Editor

PLOS ONE